# Emergent and robust ferromagnetic-insulating state in highly strained ferroelastic LaCoO$_3$ thin films

Dong Li[1,11], Hongguang Wang [2,11] ✉, Kaifeng Li[1,11], Bonan Zhu [3] ✉, Kai Jiang [4,5] ✉, Dirk Backes [6], Larissa S. I. Veiga[6], Jueli Shi[7], Pinku Roy[8,9], Ming Xiao[10], Aiping Chen [8], Quanxi Jia [9], Tien-Lin Lee[6], Sarnjeet S. Dhesi[6], David O. Scanlon [3,6], Judith L. MacManus-Driscoll [10], Peter A. van Aken [2], Kelvin H. L. Zhang [7] ✉ & Weiwei Li [1] ✉

Transition metal oxides are promising candidates for the next generation of spintronic devices due to their fascinating properties that can be effectively engineered by strain, defects, and microstructure. An excellent example can be found in ferroelastic LaCoO$_3$ with paramagnetism in bulk. In contrast, unexpected ferromagnetism is observed in tensile-strained LaCoO$_3$ films, however, its origin remains controversial. Here we simultaneously reveal the formation of ordered oxygen vacancies and previously unreported long-range suppression of CoO$_6$ octahedral rotations throughout LaCoO$_3$ films. Supported by density functional theory calculations, we find that the strong modification of Co 3$d$-O 2$p$ hybridization associated with the increase of both Co-O-Co bond angle and Co-O bond length weakens the crystal-field splitting and facilitates an ordered high-spin state of Co ions, inducing an emergent ferromagnetic-insulating state. Our work provides unique insights into underlying mechanisms driving the ferromagnetic-insulating state in tensile-strained ferroelastic LaCoO$_3$ films while suggesting potential applications toward low-power spintronic devices.

Understanding the interplay between a material's microstructure and its functionality is paramount to devising new electronic devices and technologies. This relationship is clearly evidenced by transition metal oxides (TMOs) where multiple strongly interacting degrees of freedom (spin, orbital, charge, and lattice) give rise to rich electronic phase diagrams[1]. Owing to small energy barriers among different electronic phases, external stimuli[2] such as epitaxial strain, temperature, defects, and impurities can be adopted to engineer novel functionalities and intriguing physical phenomena, e.g., insulator-superconductor transition[3], metal-insulator transition[4,5], and

[1]College of Physics, MIIT Key Laboratory of Aerospace Information Materials and Physics, State Key Laboratory of Mechanics and Control for Aerospace Structures, Nanjing University of Aeronautics and Astronautics, 211106 Nanjing, China. [2]Max Planck Institute for Solid State Research, Heisenbergstr. 1, 70569 Stuttgart, Germany. [3]Department of Chemistry, University College London, London WC1H 0AJ, UK. [4]Department of Materials, East China Normal University, 200241 Shanghai, China. [5]School of Arts and Sciences, Shanghai Dianji University, 200240 Shanghai, China. [6]Diamond Light Source Ltd., Harwell Science and Innovation Campus, Didcot, Oxfordshire OX11 0DE, UK. [7]State Key Laboratory of Physical Chemistry of Solid Surfaces, Collaborative Innovation Center of Chemistry for Energy Materials, College of Chemistry and Chemical Engineering, Xiamen University, 361005 Xiamen, China. [8]Center for Integrated Nanotechnologies (CINT), Los Alamos National Laboratory, Los Alamos, New Mexico 87545, USA. [9]Department of Materials Design and Innovation, University at Buffalo-The State University of New York, Buffalo, NY 14260, USA. [10]Department of Materials Science and Metallurgy, University of Cambridge, Cambridge CB3 0FS, UK. [11]These authors contributed equally: Dong Li, Hongguang Wang, Kaifeng Li. ✉e-mail: hgwang@fkf.mpg.de; bonan.zhu@ucl.ac.uk; kjiang@ee.ecnu.edu.cn; kelvinzhang@xmu.edu.cn; wl337@nuaa.edu.cn

magnetoelectric effects[6–8]. Among the 3$d$ TMOs family, the most remarkable property distinguishing cobalt oxides from the others is that Co ions can accommodate various spin states dictated by the competing crystal-field splitting energy ($\triangle_{CF}$), Hund's exchange energy ($\triangle_{ex}$), and $d$-orbital valence bandwidth ($W$).

Extensive attention has been paid to perovskite ferroelastic LaCoO$_3$ (LCO) recently. In bulk material, LCO exhibits a stable 3$d^6$ valence state and a rhombohedral structure ($R\bar{3}c$, $a = 5.378$ Å, $\beta = 60.81°$, which can be described by a pseudocubic lattice constant $a_{pc} = 3.8029$ Å)[9,10]. Also, it displays an insulating behavior and has no long-range magnetic ordering. With increasing temperature, the Co$^{3+}$ ions show an active spin-state transition, from a low spin (LS, $t_{2g}^6$, $S = 0$) to an intermediate spin (IS, $t_{2g}^5 e_g^1$, $S = 1$) or a high spin (HS, $t_{2g}^4 e_g^2$, $S = 2$). By annealing a compressive-strained LCO epitaxial film in vacuum, the LaCoO$_{2.5}$ phase with a *zigzag*-like oxygen vacancies ($V_O$) ordering shows a ferromagnetism (FM) insulating state[11]. In addition, numerous studies reported an unexpected FM with a Curie temperature ($T_C$) of 80–85 K observed in the tensile-strained LCO epitaxial films[12–18]. The underlying physical mechanism of emergent FM and spin-state transition remains highly controversial. Previously, the tensile-strain-induced ferroelastic distortion was proposed as the driving force for inducing FM order[14,17,18]. The tensile strain was partially relaxed by the formation of a superstructure consisting of bright and dark stripes in scanning transmission electron microscopy (STEM) images.

Since the local structure is strongly correlated with electronic structures and magnetic properties, precise knowledge of the local structure is required to understand the nature of FM. However, the local atomic structure resulting from ferroelastic distortion is not well explained so far. One mechanism has been proposed that attributes FM to an ordered array of $V_O$ formed in dark stripes observed in the STEM images[15,16]. Due to the formation of $V_O$, the valence state of Co$^{3+}$ is changed to Co$^{2+}$ with a HS electronic configuration ($t_{2g}^5 e_g^2$, $S = 3/2$), thus favoring long-range FM order. O $K$-edge electron-energy-loss spectroscopy (EELS) spectra captured from bright and dark stripes showed that the intensity of a pre-peak feature at an energy of 527–528 eV (Co 3$d$–O 2$p$ hybridization) is strongly suppressed in dark stripes in comparison to bright stripes, suggesting the presence of $V_O$. However, the existence of Co$^{2+}$ in dark stripes has been heavily questioned, because the Co $L_{2,3}$-edge EELS spectra showed no visible change of peak position between bright and dark stripes[15]. In addition, there is not much difference in the Co $L_3/L_2$ ratio in the dark and bright stripes[15]. Density functional theory (DFT) calculations have also been applied to investigate the nature of FM in the tensile-strained LCO[19,20]. Importantly, it was found that the tensile strain-induced changes in lattice constants are insufficient to stabilize long-range FM order[20]. The suppression of CoO$_6$ octahedral rotations should be considered to modify $e_g$ orbital ordering configuration, inducing a spin-state transition to an FM state. Yet, until now, direct experimental evidence to support the theoretical prediction has not been reported in the literature. To summarize, there is an agreement, both experimentally and theoretically, that the redistribution of Co orbital occupation occurs under tensile strain in LCO epitaxial films, producing macroscopic FM. However, the origin of FM order remains highly elusive. The underlying correlation between the electronic structures, spin states, and local atomic structure has not yet been fully understood.

In this work, we systematically investigate the atomic and electronic structures, and their correlation with the spin-state transition and the FM insulating state of high-quality LCO epitaxial films by using the measurements of STEM, X-ray photoelectron spectroscopy (XPS), and X-ray absorption spectroscopy (XAS). We provide direct evidence for confirming the formation of ordered $V_O$ in dark stripes observed in the STEM images and further reveal previously unreported long-range suppression of CoO$_6$ octahedral rotations throughout LCO epitaxial films. As a consequence, this strongly modifies Co orbital occupancy,

Co-O-Co bond angle, and Co-O bond length, which weakens the crystal-field splitting and facilitates an ordered high-spin state of Co ions. DFT calculations further unravel the underlying physical mechanism of spin-state transition for inducing an emergent and robust FM insulating state. Overall, our work provides new insights into the mechanisms driving the FM insulating state in tensile-strained ferroelastic LCO epitaxial films.

## Results

### Microstructural investigations and physical properties of LCO films

LCO epitaxial films with a thickness ranging from 4 to 25 unit cells (uc) were fabricated by pulsed laser deposition (PLD). Distinct thickness fringes around the Bragg peaks were observed in X-ray diffraction (XRD) $\theta$–$2\theta$ scans (Supplementary Fig. 1a), demonstrating the high crystalline quality of LCO films. Ex-situ atomic force microscopy (AFM) images confirmed that all LCO films have an atomically smooth surface with a well-defined terrace structure (Supplementary Fig. 1b). Reciprocal space maps of (103) Bragg reflections of STO and LCO exhibited that LCO thin films are fully tensile strained to the STO substrates along the in-plane direction (Supplementary Fig. 2). Magnetic measurements showed a magnetic hysteresis loop and a $T_C$ of ~85 K in 25-uc-thick LCO film (Fig. 1a, b), consistent with previous works[12–17]. Hence, this means long-range FM order formed in our LCO films. XAS (Supplementary Fig. 3a) and X-ray magnetic circular dichroism (XMCD) (Supplementary Fig. 3b) measurements at Co-$L_{2,3}$ edges further revealed that FM originates from Co ion in the film and the change in the spin states of Co ions in the films is consistent with the formation of ordered $V_O$[16]. Note that the magnetization and the $T_C$ values of LCO films decrease with decreasing film thickness (Fig. 1a, b). In addition, an insulating behavior is observed in 25-uc-thick LCO film by probing the temperature-dependent resistivity curve (Supplementary Fig. 4), suggesting the film has FM insulating behavior.

### Microstructure and electronic structure of LCO films by STEM and EELS

The microstructures of the LCO films with thicknesses of 6 uc and 25 uc were further investigated using high-angle annular dark-field (HAADF) STEM[8,21,22] (Fig. 1c–i, Supplementary Figs. 5 and 6a). EELS elemental maps for La-$M_{4,5}$, Sr-$L_{2,3}$, Co-$L_{2,3}$, and Ti-$L_{2,3}$ edges clearly validate the uniform distribution of elements within the films (Fig. 1d–h and Supplementary Fig. 6b–f). Figure 1i displays the EELS line profiles of individual elements of 25-uc-thick LCO film, revealing an abrupt interface between LCO and STO without the formation of any misfit dislocations (Supplementary Fig. 7). The interfacial intermixing is limited within 1 uc and the termination is identified to be TiO$_2$-LaO at the interface. More importantly, well-ordered dark stripes with a period ~3 uc were distinctly observed in the interior part of the LCO film (Fig. 1c), agreeing well with the observations in the previous reports[14–16,23,24]. Note that the interfacial LCO layer (~3 uc) close to the STO substrate is free from the dark stripes (Supplementary Fig. 5). We also found that no dark stripes are observed in 6-uc-thick LCO film (Supplementary Fig. 6a). These results strongly imply that dark stripes are gradually formed in LCO film with increasing thickness. Furthermore, the gradual evolution of dark stripes occurs simultaneously with emergent and robust FM in LCO film with thickness, suggesting a link between dark stripes and FM.

To determine whether the observed dark stripes are associated with the formation of $V_O$, we carefully investigated atomic-scale electronic structures of the LCO films using EELS measurements (Fig. 2a). An EELS line scan measured from an atomic-resolution HAADF-STEM image of 25-uc-thick LCO film is shown in Fig. 2b, and a periodic reduction of the intensity of the Co-$L_{2,3}$ edges is observed, coinciding with the positions of observed dark stripes. Figure 2c shows EELS spectra of the O $K$-edge extracted from line scans acquired along

bright and dark stripes. Three peaks were distinctly observed and are labeled as "A", "B", and "C". The pre-peak marked as "A" can be assigned to unoccupied Co $3d$ hybridized with O $2p$, whereas peaks "B" and "C" originate from the hybridization of O $2p$ with La $5d$ and Co 4sp, respectively[25]. Compared with bright stripes, the intensity of these three peaks in dark stripes is significantly reduced. More importantly, the pre-peak "A" in dark stripes almost disappears. Previously, the suppressed pre-peak of O $K$-edge in LCO was attributed to the formation of $V_O$[15,16] or to the presence of different spin states[26]. To further identify the origin of the suppressed pre-peak "A" in dark stripes, we turned to measure the Co-$L_{2,3}$ edge EELS spectra, which are very sensitive to the oxidation state of Co atom[25,27]. As shown in Fig. 2d, in comparison to bright stripes, the peak positions of Co-$L_{2,3}$ edges in dark stripes clearly shift toward lower energy around 0.8–0.9 eV, indicating the reduction of the Co oxidation state. Previous works[27,28] reported that the Co oxidation state decreases with the increase of the Co $L_3/L_2$ ratio. By comparing the Co $L_3/L_2$ ratio of bright and dark stripes with the value obtained from literatures[27,28], we found that this ratio is increased in dark stripes and further identified that the oxidation states of Co in bright stripes and dark stripes are around +2.85 and +2.15 (Fig. 2g), respectively. These results undoubtedly confirm that the origin of dark stripes observed in 25-uc-thick LCO film arises from the formation of ordered $V_O$. However, in stark contrast to the observation in 25-uc-thick LCO film, due to the fact that no dark stripes are formed in 6-uc-thick LCO film, it was found that no visible change in EELS spectra of O $K$-edge and Co-$L_{2,3}$ edges can be detected (Fig. 2e, f and Supplementary Fig. 8a–c). These results strongly imply that the formation of ordered $V_O$ is gradually developed in LCO film with increasing thickness, consistent with a previous report[29], and further

corroborating that in 25-uc-thick LCO film, the formation of dark stripes or ordered $V_O$ is strongly coupled with emergent FM order.

Annular bright-field (ABF) STEM imaging[8,22,30] was further applied to quantitatively measure the cation and anion positions (Supplementary Figs. 9 and 10) for determining the Co-O-Co bond angle ($\beta_{Co-O-Co}$) and the Co-O bond length ($d_{Co-O}$). Figure 3 (top panel) shows representative ABF-STEM images of 6-uc-thick and 25-uc-thick LCO films, respectively. According to the Glazer notation[31,32], the TiO$_6$ rotation pattern in the STO substrate is confirmed as $a^0a^0a^0$ (Supplementary Figs. 9a–c and 10a) and the rotation pattern of LCO is determined to be $a^-a^-c^-$ or $a^-a^-c^+$ (Supplementary Fig. 10b), consistent with $a^-a^-a^-$ of bulk LCO[20]. From neutron diffraction measurements at room temperature[10], $\beta_{Co-O-Co}$ and $d_{Co-O}$ were determined to be -163.5° and -1.927 Å for bulk LCO, respectively. The plane-averaged and projected tilt angles across LCO/STO interface were determined from ABF-STEM images and shown in the bottom panel of Fig. 3. Due to different symmetry between STO (cubic) and LCO (rhombohedral), we found that the $\beta_{Co-O-Co}$ is increased from bulk 163.5°[10,33,34] to 176.7° in 6-uc-thick LCO by the strong suppression of CoO$_6$ rotations. Previous works reported that the impact length scale of symmetry-induced modification of BO$_6$ octahedral rotations is only 2–6 uc confined at the interface region and its modulation decays very quickly away from the interface[35–37]. Surprisingly, we found that the suppressed CoO$_6$ rotations in 25-uc-thick LCO film are very robust and can be observed throughout of the film. As a consequence, the $\beta_{Co-O-Co}$ is increased from bulk 163.5° to 172.5°, which is only slightly smaller than the value of 176.7° in 6-uc-thick LCO. It should be emphasized that this observation has never been reported in previous studies. We speculate that it may be caused by the formation of ordered $V_O$ or dark stripes

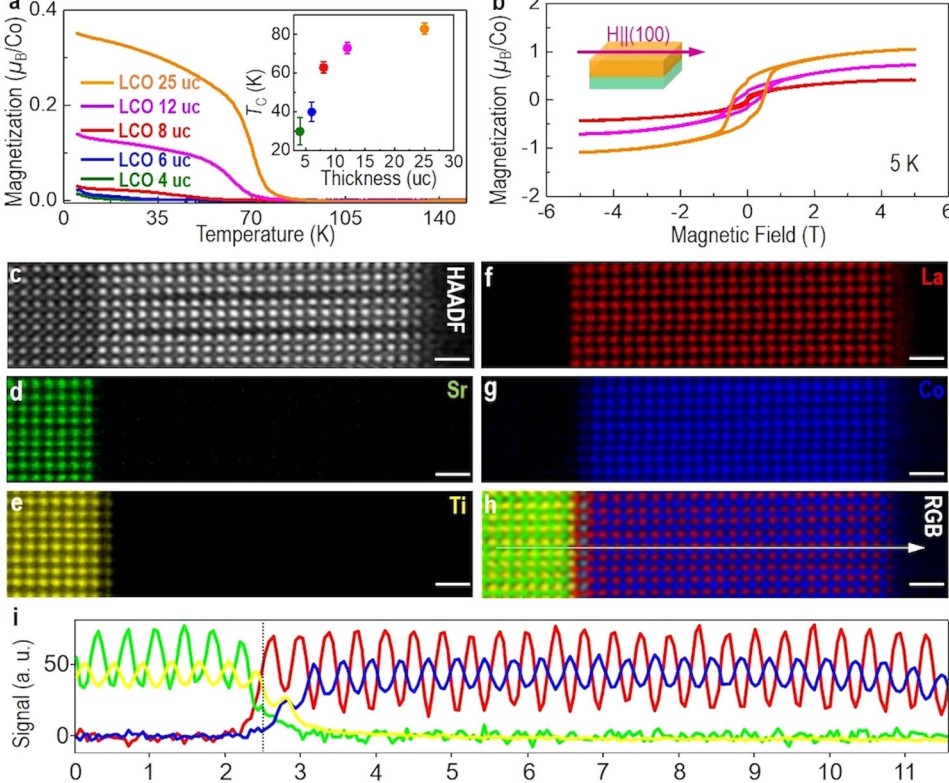

**Fig. 1 | Magnetic properties and STEM investigations of LCO films.**
**a** Temperature and **b** magnetic field-dependent magnetization of LCO thin films with different thicknesses measured by applying a magnetic field along the IP direction. The inset of (**a**) shows the thickness-dependent FM $T_C$. Temperature-dependent magnetization curves were measured during LCO thin films warming after a field cooling of 1 kOe. **c** Z-contrast HAADF-STEM image of LCO 25 uc in cross-sectional orientation, projected along the [010] zone axis. **d**–**h** Atomically resolved STEM-EELS elemental maps of Sr, Ti, La, and Co, and a colored RGB overlay, respectively. Scale bar: 1 nm. **i** EELS signal intensity profiles across the LCO/STO interface extracted along the white arrow direction in (**h**).

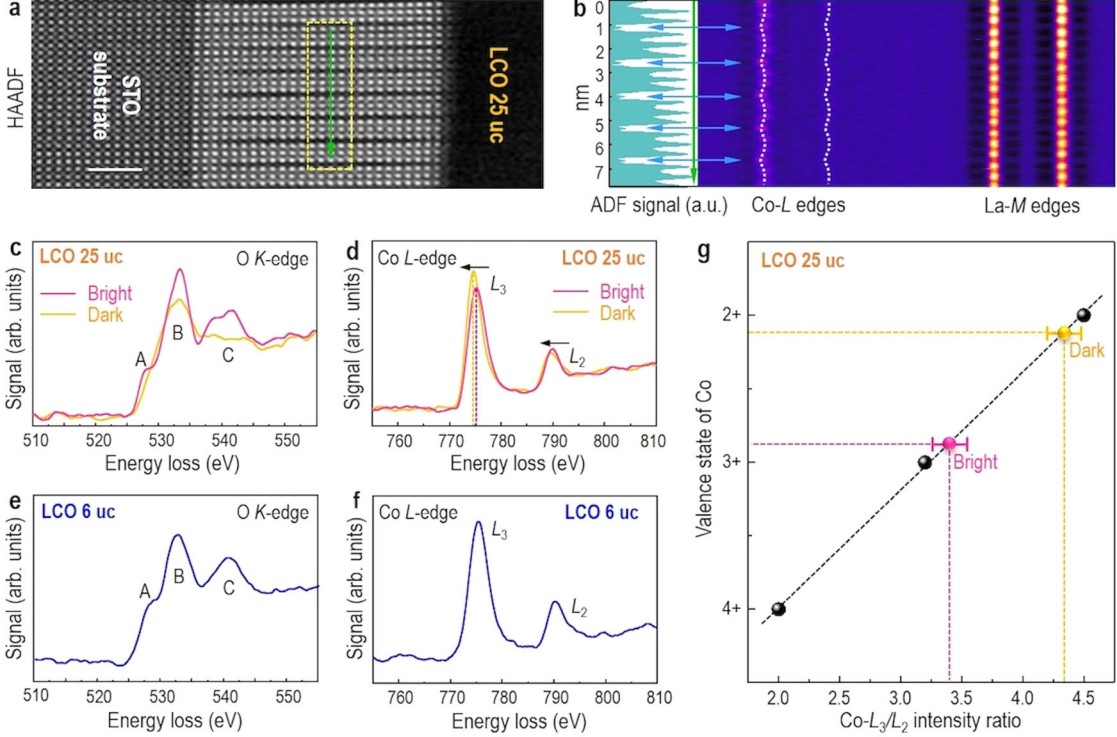

**Fig. 2 | EELS fine structures of LCO films. a** Z-contrast HAADF-STEM image of LCO 25 uc in cross-sectional orientation, projected along the [010] zone axis. Scale bar: 2 nm. **b** The second derivative of the EELS line-scan along the green arrow direction, highlighting the variation of the peak position of Co-$L_{2,3}$ edges. O $K$-edge EELS spectra of **c** LCO 25 uc and **e** LCO 6 uc. Co $L_{2,3}$-edge EELS spectra of **d** LCO 25 uc and **f** LCO 6 uc. **g** Calculated Co $L_3/L_2$ intensity ratios and corresponding valence states of Co for the bright and dark stripes in comparison to literature values of Co$^{2+}$, Co$^{3+}$, and Co$^{4+}$ [27,28] (black points).

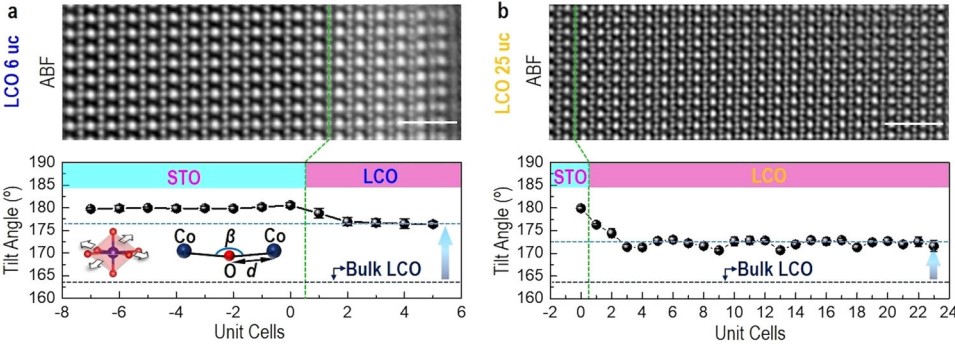

**Fig. 3 | Bond angle analysis of LCO films.** ABF-STEM images of LCO 6 uc in the top panel of (**a**) and LCO 25 uc in the top panel of (**b**). Plane-averaged Co-O-Co tilt angles of LCO 6 uc in the bottom panel of (**a**) and LCO 25 uc in the bottom panel of (**b**). Scale bar: 1 nm.

throughout the film. Owing to LCO films being fully tensed by STO substrates along in-plane (IP) direction, an average IP Co-O bond length ($d_{Co-O}$) was further determined to be around 1.952 Å for both 6-uc-thick and 25-uc-thick LCO films, which is larger than the bulk bond length ($d_{Co-O}$ ~ 1.927 Å)[10,33,34]. Furthermore, the $V_O$-induced lattice expansion was observed along the dark stripes in 25-uc-thick LCO film (Supplementary Fig. 11a–c), in comparison to 6-uc-thick LCO film (Supplementary Fig. 11d–f).

## Electronic structure of LCO films by DFT calculations

To gain insight into the correlation between local atomic structure and electronic structures, we performed constrained ab initio Random Structure Searching (AIRSS)[38,39] to seek low-energy configurations. The positions of the cations and the unit cell sizes are fixed when generating the random structures as they can be determined based on

STEM images, but they are allowed to move for structural relaxation. The searches are performed with unit cells of La$_6$Co$_8$O$_{18-x}$, with $x$ ranging from 0 to 4. The lowest energy structure with $x = 2$ is found to give the best fit to the HAADF-STEM images and reproduces the increased La-La distances, in good agreement with the experimental observation of $V_O$ in dark stripes and IP lattice constant of 3.905 Å. This structure resembles the previously reported theoretical model[14]. Due to the unit cell periodicity, the existence of tetrahedral coordinated Co results in an anti-phase boundary of the CoO$_6$ rotation network. If such a boundary does exist, the rotations would not be visible along the [110] direction, which is not consistent with the experimental observation. Hence, a revised structural model with increased periodicity along the $x$ direction is proposed, allowing the CoO$_6$ rotation network to carry on through the plane of ordered $V_O$ (Fig. 4a). However, relaxing this new model returns the rotation network to the initial

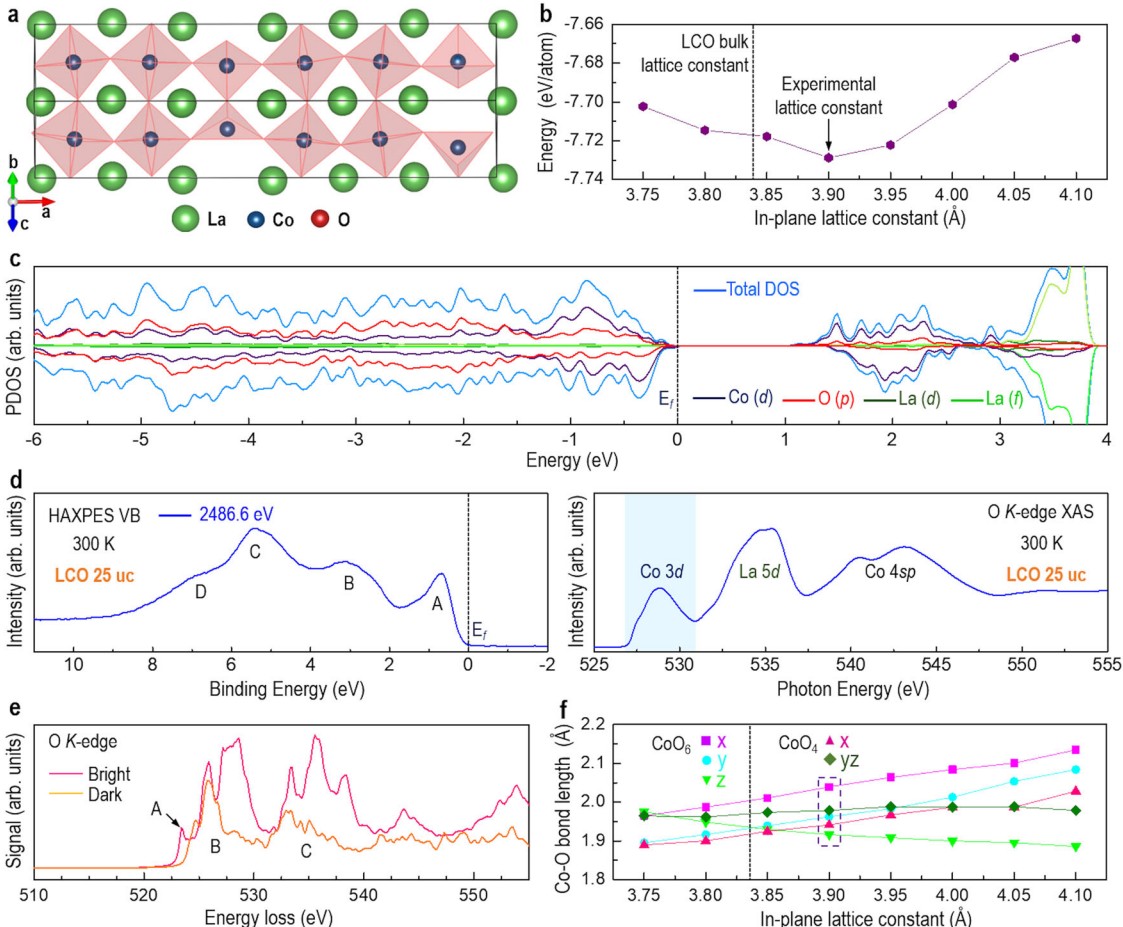

**Fig. 4 | Electronic structure and bond length of LCO film. a** Schematic view of LCO with ordered $V_O$. **b** Energies of the relaxed structures containing ordered $V_O$ with different in-plane lattice constant. **c** Projected density of state (PDOS) spectrum for LCO with ordered $V_O$ or dark stripes. **d** XPS valence band (VB) spectrum excited with a photon energy of 2486.6 eV (left panel) and O $K$-edge XAS spectrum (right panel) for LCO 25 uc. **e** Theoretical O $K$-edge EELS spectra obtained from bright and dark stripes. **f** The modulation of Co-O bond length as a function of lattice constant.

setting, which indicates a strong pinning effect on the octahedral network due to the plane of ordered $V_O$. Geometry optimization with constrained lattice vectors along the $x$ and $y$ directions shows that the inclusion of $V_O$ increases the effective lattice constant beyond that of the bulk value (Fig. 4b), suggesting that the tensile strain acts as an additional driving force for $V_O$ formation. The tetrahedral coordinated Co atoms are displaced from their ideal octahedral positions, resulting in a *zigzag* arrangement which makes the column of atoms no longer aligned when viewed along the same direction as in Fig. 4a. We note that for clarity, Fig. 4a only includes a single layer of LCO which does not show this effect. Such misaligned atomic column could also be one possible origin of the reduced contrast in the dark region of the HAADF image (Fig. 1c). Note that although $V_O$ introduces the electron to the LCO, an insulating behavior was confirmed by both the total density of states (DOS) result (Fig. 4c) and the experiment (Supplementary Fig. 4), consistent with the previous reports[14,15,17]. The HS-LS checkerboard spin configuration in the pristine LCO without ordered $V_O$ was found to have the lowest energy with a band gap opening between the $t_{2g}$ states of HS and LS $Co^{3+}$ (Supplementary Fig. 12). The presence of $V_O$ would result in conducting states as a result of the extra electrons. This is not the case here as the extra electrons lead to the reduction of $Co^{3+}$ to $Co^{2+}$, while the nature of the band gap remains unchanged, which is between the Co $3d$ states caused by the crystal-field splitting. Additionally, the theoretical calculated bandgap (-0.925 eV, Fig. 4c) is comparable to the experimental value (-0.840 eV)[40]. To further understand the occupied and unoccupied DOS near the Fermi level

$(E_f)$[24,41–43], the measured valence band (VB) spectrum and XAS at the O $K$-edge are presented in Fig. 4d. In comparison to previous reports[42,43], we determined that peak A originates from occupied Co $3d$ states, peak B is attributed to the dominant O $2p$ states hybridized with minor occupied Co $3d$ states, and peak C and shoulder D have some occupied Co $3d$ contributions. O $K$-edge XAS spectrum shown in the right panel of Fig. 4d probes excitation of O $1s$ electron to O $2p$ states hybridized with unoccupied Co $3d$ states, La $5d$ states, and Co $4sp$ states above $E_f$. The overall shapes of our measured O $K$-edge XAS spectrum agree well with previously reported spectra[24,42], with the peak at 528.9 eV assigned to unoccupied Co $3d$ states hybridized with O $2p$ states. More importantly, experimental DOS near $E_f$ is in good agreement with the DFT calculated electronic structure (Fig. 4c).

Figure 4e shows the theoretically calculated EELS spectra of O $K$-edge extracted from bright and dark stripes. Three peaks labeled "A", "B", and "C" were distinctly observed, consistent with the experimental data (Fig. 2c). Compared with bright stripes, these three peaks in dark stripes show a substantial reduction in their intensity. Obviously, the pre-peak "A" in dark stripes is strongly suppressed and disappears, agreeing very well with the experimental observation (Fig. 2c). This again supports that ordered $V_O$ is formed in dark stripes in 25-uc-thick LCO film. In addition, at the lowest energy structure with an IP lattice constant of 3.90 Å (Fig. 4f), the IP $d_{Co-O}$ of the $CoO_6$ octahedra along the $x$ and $y$ axis is around 2.04 Å and 1.96 Å, respectively, and the IP $d_{Co-O}$ of the $CoO_4$ tetrahedra along $x$ is around 1.94 Å. We found that, under the same experimental tensile strain, the theoretical predicted

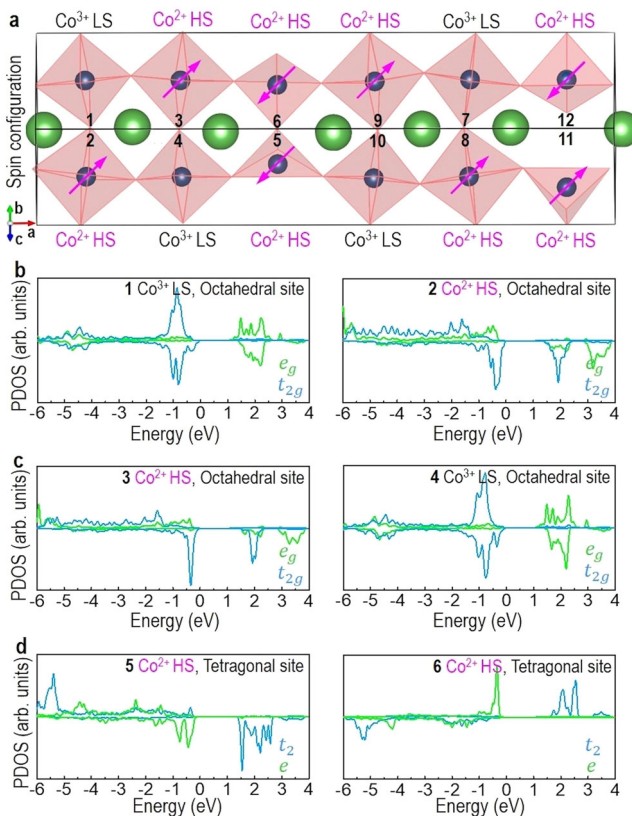

**Fig. 5 | Spin configurations and PDOS spectra of LCO film, from DFT calculations. a** Schematic model of the structure, oxidation states, and spin states. **b–d** PDOS projected over different Co ions in octahedral and tetragonal sites.

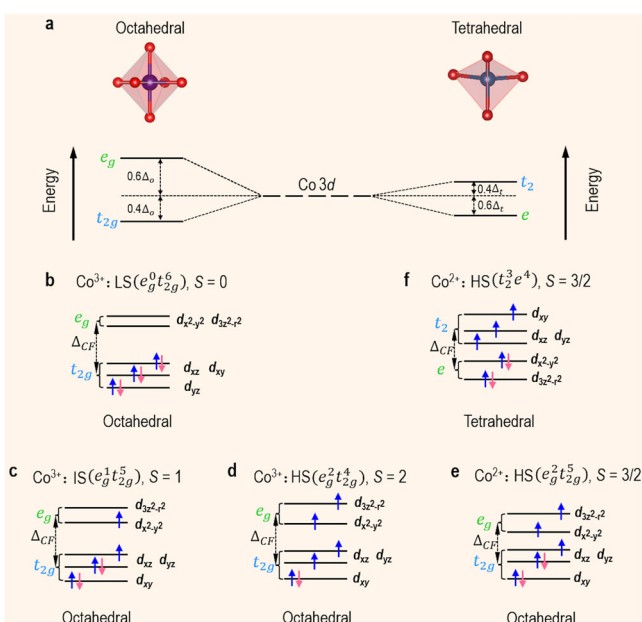

**Fig. 6 | Electronic structure of Co 3*d* orbital state. a** Schematic crystal-field splitting of Co 3*d* orbitals in octahedral and tetrahedral coordination. Schematic energy-level diagrams of **b–e** a $Co^{3+}$ ion with LS, IS, and HS configurations in octahedral coordination, and **f** a $Co^{2+}$ ion with HS configuration in tetrahedral coordination.

IP $d_{Co-O}$ is larger than bulk $d_{Co-O}$ (1.927 Å)[33,34], in good agreement with the experimental results.

## Discussion

We further investigated the spin configuration and magnetic spin state of LCO with ordered $V_O$ (or dark stripes) as shown in Fig. 5. Eight $Co^{2+}$ are identified based on the projected magnetization, which is consistent with having four $V_O$ in the unit cell. The tetrahedral coordinated Co atoms can be assigned as HS $Co^{2+}$. Octahedral coordinated HS $Co^{2+}$ and LS $Co^{3+}$ atoms are in a checkerboard arrangement. In Fig. 5a, the $CoO_6$ octahedra with $Co^{2+}$ have an increased volume compared to those of $Co^{3+}$, which is consistent with the former having larger ionic radii and smaller electrostatic attraction. Figure 5b–d shows the calculated DOS projected over different Co ions. We found that the Co $t_{2g}$ states are fully filled whereas the Co $e_g$ states are empty in the LS $Co^{3+}$ octahedral sites (1 and 4). The occupied Co $e_g$ states are increased in the HS $Co^{2+}$ octahedral sites (2 and 3). For tetrahedral coordinated $Co^{2+}$ (5 and 6) the $e$ states are completely occupied and the conduction band is made of unoccupied $t_2$ states.

Our theoretical and experimental results together demonstrate that an emergent FM insulating state in the LCO films is strongly correlated with the formation of ordered $V_O$ in dark stripes and the suppression of octahedral rotations. The energy for realizing spin-state transition is determined by $\triangle = \triangle_{CF} - \triangle_{ex} - W/2$. $\triangle_{ex}$ is an intrinsic material constant, whereas both $\triangle_{CF} (\propto d_{Co-O}^{-5})$ and $W (\propto d_{Co-O}^{-3.5} \sin(\beta/2))$ are strongly dependent on $d_{Co-O}$ and $\beta_{Co-O-Co}$. Therefore, the modification of $d_{Co-O}$ and $\beta_{Co-O-Co}$ directly influences the delicate balance between $\triangle_{CF}$ and $W$, thereby controlling the spin-state transition. Figure 6 shows a schematic of the electronic structures and spin states as a response to the modification in $\triangle_{CF}$ and $W$, i.e.,

$d_{Co-O}$ and $\beta_{Co-O-Co}$. Applying epitaxial tensile strain and introducing ordered $V_O$ significantly changes the distortion (Fig. 6a), thereby reducing orbital degeneracy. In addition, based on the analysis of ABF-STEM images (Fig. 3), we identified the increase of both $d_{Co-O}$ and $\beta_{Co-O-Co}$ in 25-uc-thick LCO film. This will dramatically decrease $\triangle_{CF}$, increase $W$, and further reduce the gap between $e_g$ ($e$) and $t_{2g}$ ($t_2$) levels as favored by Hund's rules[44]. In addition, owing to the LCO films fully tensed by the STO substrates along the IP direction, the energy of $d_{x^2-y^2}$ orbital is lower than that of $d_{3z^2-r^2}$ orbital (Supplementary Fig. 13). This means electrons excited from $t_{2g}$ bands preferentially occupy the $d_{x^2-y^2}$ orbital rather than $d_{3z^2-r^2}$ orbital in Co $e_g$ bands. Consequently, the increased population of $e_g$ or $t_2$ electrons in 25-uc-thick LCO film can favor the transition of the $Co^{3+}$ from an LS (Fig. 6b) to an HS state (Fig. 6c–f). DFT calculations demonstrated that the occupied $e_g$ or $t_2$ states are from the HS $Co^{2+}$ octahedral sites and the HS $Co^{2+}$ tetragonal sites (i.e., dark stripes or $V_O$) (Fig. 5b–d). The checkerboard arrangement of octahedral HS $Co^{2+}$ and octahedral LS $Co^{3+}$ inhibits super-exchange interactions among octahedral HS $Co^{2+}$. DFT calculations found many possible spin arrangements among the $Co^{2+}$ coupled with the alignment of the checkerboard $Co^{3+}$ arrangement through the planes of $V_O$. Considering a repeating unit cells of four Co ions in octahedral coordination and two Co ions in tetrahedral coordinate, the only possible total spins are 12 $\mu_B$, 6 $\mu_B$, and 0 $\mu_B$. The fully FM case with HS $Co^{3+}$ (total spin 20 $\mu_B$) is found to result in much higher energy, up to more than 50 meV per atom, while multiple arrangements with a total spin of 12 $\mu_B$, 6 $\mu_B$ or 0 $\mu_B$ are found to give lower energies. Due to the small difference in energies (in the order of a few meV per atom), a unique ground state spin arrangement cannot be determined reliably using DFT. Figure 5a shows one of the low-energy spin arrangements with a total spin of 0 $\mu_B$ among Co1-Co6 and a total spin of 6 $\mu_B$ among Co7-Co12, resulting in an average spin of 0.5 $\mu_B$/Co. Alternative arrangements can have a total spin of 6 $\mu_B$ among Co1-Co6, giving an average spin of 1.0 $\mu_B$/Co. This is consistent with the experimentally observed value of 1.05 $\mu_B$/Co (Fig. 1b). Hence, we attribute the origin of the observed emergent long-range FM order to the super-exchange interactions of Co-O-Co, the arrangement of octahedral and

tetrahedral $Co^{2+}$, and enhanced $e_g/t_2$ occupation, which is induced by the formation of ordered $V_O$ and the suppression of octahedral rotations.

In summary, we have combined multiple experimental probes and theoretical calculations to determine the origin of unexpected FM insulating state in tensile-strained ferroelastic LCO epitaxial films. Dark stripes observed in STEM images are unambiguously assigned to the formation of ordered $V_O$, reducing oxidation states of Co from +2.85 to +2.15 and changing the Co 3$d$-O 2$p$ hybridization. More importantly, we reveal previously unreported long-range suppressed $CoO_6$ octahedral rotations throughout 25-uc-thick LCO film, inducing an increase of $\beta_{Co-O-Co}$ from bulk 163.5° to 172.5° and IP $d_{Co-O}$ from bulk 1.927 Å to 1.952 Å simultaneously. Consequently, owing to the dual effect of the formation of ordered $V_O$ and long-range suppressed $CoO_6$ octahedral rotations, the crystal-field splitting is weakened, thus promoting the ordered high-spin state of Co ions, and producing an emergent and robust ferromagnetic-insulating state. This work presents new insights into the understanding of how external parameters lead to a delicate competition between crystal-field splitting energy, Hund's exchange energy, and $d$-orbital valence bandwidth, inducing emergent multi-ferroic functionalities in cobalt-based oxide epitaxial films, and offers new opportunities for creating low-power consumption spintronic devices.

## Methods

### Thin film preparation
High-quality $LaCoO_3$ (LCO) epitaxial thin films were epitaxially grown on (001) $TiO_2$-terminated $SrTiO_3$ (STO) substrates. Films were grown at a temperature of 665 °C in an oxygen partial pressure of 100 mTorr, where the pulsed laser deposition was carried out by a KrF excimer laser (248 nm) at a laser fluence of 1.5 J cm$^{-2}$ and 2 Hz. During the growth, in-situ reflection high-energy electron-diffraction (RHEED) intensity oscillations were monitored to control the thickness of the LCO layer of different unit cells (uc). After growth, the films were cooled down to room temperature under an oxygen pressure of 200 Torr.

### Structural characterizations
The crystal structures and surface morphologies of the films were characterized by high-resolution X-ray diffraction (XRD) with Cu K$_\alpha$ radiation ($\lambda = 1.5405$ Å) (Empyrean, PANalytical) and atomic force microscopy (AFM), respectively.

### Physical properties characterizations
Macroscopic magnetic measurements were performed with an MPMS3 SQUID-VSM magnetometer (Quantum Design). The magnetic field was applied parallel to the film plane during the measurements. Transport measurements were measured by a Quantum Design Physical Property Measurement System (PPMS).

### Scanning transmission electron microscopy
Cross-sectional specimens oriented along the $[100]_{pc}$ and $[110]_{pc}$ direction for STEM analysis were prepared by conventional mechanical thinning, precision polishing, and ion milling[45]. STEM studies were conducted using a spherical aberration-corrected STEM (JEM-ARM200F, JEOL Co. Ltd.) equipped with a cold-field emission gun and a DCOR probe Cs-corrector (CEOS GmbH) operated at 200 kV. The STEM images were obtained by an ADF detector with a convergent semi-angle of 20.4 mrad and collection semi-angles of 70–300 mrad. In order to make precise measurements of lattice constants, ten serial frames were acquired with a short dwell time (2 μs/pixel), aligned, and added afterward to improve the signal-to-noise ratio (SNR) and to minimize the image distortion of HAADF images. Atomically resolved HAADF-STEM images have been analyzed using Geometric Phase Analysis (GPA) for strain analysis. EELS acquisition has been performed with a Gatan GIF Quantum ERS imaging filter equipped with a Gatan K2

Summit camera and a CCD camera with a convergent semi-angle of 20.4 mrad and a collection semi-angle of 111 mrad. EELS spectrum imaging was performed with a dispersion of 0.5 eV/channel for the simultaneous acquisition of signals of the Ti-$L_{2,3}$, O-$K$, Co-$L_{2,3}$, La-$M_{4,5}$, and Sr-$L_{2,3}$ edges. EELS line scans were conducted with a dispersion of 0.1 eV/channel using a CCD camera in the Dual-EELS mode for further fine structure analysis of the O-$K$ and Co-$L_{2,3}$ edges. The corresponding energy resolution is about 0.6 eV (Supplementary Fig. 14). The raw spectrum image data were denoised by applying a principal component analysis (PCA) with the multivariate statistical analysis (MSA) plugin (HREM Research Inc.) in Gatan Digital Micrograph. The coordinate positions of oxygen ions were determined by Gaussian fitting the oxygen columns in ABF images and then used to calculate spacing between O ions and the angles between neighboring $CoO_6$ octahedra[46,47]. The effect of electron dose conditions on EELS measurements was evaluated according to the literature[48], showing that the observed red shift of Co-$L_3$ edge in dark stripes compared to bright stripes is not caused by the electron dose (Supplementary Fig. 15).

### Spectroscopic measurements
X-ray photoelectron spectroscopy (XPS) measurements were performed using 2486.6 eV photon energy at the I09 beamline of the Diamond Light Source. XPS spectrum was energy-resolved and measured using a VG Scienta EW4000 high-energy analyzer. The Fermi level of the LCO films was calibrated using a polycrystalline Au foil. XAS measurements with polarization dependence were performed at the I06 beamline of the Diamond Light Source. X-ray magnetic circular dichroism (XMCD) measurements were carried out by probing the total electron yield (TEY) in grazing incidence geometry. XMCD measurements were performed at 2 K under a 6 T magnetic field applied along the a-b plane of the films, parallel to the beam propagation direction. To ensure that the XMCD signal is of magnetic origin, the magnetic field was also applied in the opposite direction to verify that the sign of the XMCD reversed. X-ray linear dichroism (XLD) measurements were carried out at 300 K without a magnetic field while the TEY was detected in grazing incidence geometry. The XLD spectra were obtained by the intensity difference ($I_v$-$I_h$) between the spectra measured with horizontal ($E_h$) and vertical ($E_v$) linear polarizations.

### Density functional theory
Plane-wave pseudopotential density functional theory (DFT) calculations were carried out using the Vienna Ab initio Simulation Package (VASP)[49,50] with a plane-wave cutoff energy of 550 eV, gamma centered K point grids with reciprocal space spacing 0.314 Å$^{-1}$ and the PBEsol[51] exchange-correlation functional and Hubbard U of 3 eV applied to the Co $d$ orbitals. The O, Co, and La pseudopotentials in the PBE_54 dataset are used. The CASTEP[52] plane-wave DFT code is used for structure searching with on-the-fly generated core-corrected ultrasoft pseudopotentials from the QC5 library and a cutoff energy of 340 eV. The K point spacing during the search is set to 0.440 Å$^{-1}$. The AiiDA[53,54] framework is used to manage the calculations and record their provenance. The atomistic simulation environment is used for building structural models[55].

### Reporting summary
Further information on research design is available in the Nature Portfolio Reporting Summary linked to this article.

## Data availability
All the experimental/calculation data that support the findings of this study are available from the corresponding authors upon request.

## Code availability
All the codes used for this study are available from the corresponding authors upon request.

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

## Acknowledgements

W.-W.L. and K.H.L.Z. acknowledge support from the National Natural Science Foundation of China (Grant No. 52102177, Grant No. 21872116, and No. 22075232). W.-W.L. is grateful for funding support from the National Natural Science Foundation of Jiangsu Province (Grant No. BK20210313), the Top-notch Academic Programs Project of Jiangsu Higher Education Institutions (TAPP), and the Jiangsu Specially-Appointed Professor Program. The research work is supported by the supporting funds for the talents of Nanjing University of Aeronautics and Astronautics. We thank the Diamond Light Source for the time on beamline I06 (proposal MM25425, MM26901, and MM29616) and beamline I09 (proposal SI31069). J.L.M.-D. and M.X. thank the Royal Academy of Engineering, grant CIET1819_24, for funding. Via our membership of the UK's HEC Materials Chemistry Consortium, which is funded by EPSRC (EP/R029431), this work used the ARCHER2 UK National Supercomputing Service (http://www.archer2.ac.uk), the UK Materials and Molecular Modelling Hub for computational resources, MMM Hub, which is partially funded by EPSRC (EP/P020194). The authors also acknowledge the use of the UCL Myriad and Kathleen High Performance Computing Facility (Myriad@UCL, Kathleen@UCL), and associated support services, in the completion of this work. The work at Los Alamos National Laboratory was supported by the NNSA's Laboratory Directed Research and Development Program and was performed, in part, at the Center for Integrated Nanotechnologies (CINT), an Office of Science User Facility operated for the U.S. Department of Energy Office of Science. Los Alamos National Laboratory, an affirmative action equal opportunity employer, is managed by Triad National Security, LLC for the U.S. Department of Energy's NNSA, under contract 89233218CNA000001. The work at the University at Buffalo was partially supported by the U.S. National Science Foundation under award number ECCS-1902623. Q.X.J. also acknowledges the CINT Users Program. This project has received funding from the European Union's Horizon 2020 research and innovation programme under Grant Agreement No. 823717-ESTEEM3.

## Author contributions

W.-W.L. and H.G.W. conceived the project and directed the research. D.L., H.G.W., and K.L. contributed equally. D.L. and K.L. did sample preparation and performed XRD and magnetic measurements with the help of K.J., P.R., M.X., A.C., Q.X.J., and J.L.M.D. W.-W.L., D.B., L.V., and S.D. performed XLD and XMCD measurements. J.S., T.L., and K.H.L.Z. carried out XPS and transport measurements. H.G.W. and P.A.v.A. conducted STEM measurements. B.Z. and D.S. performed DFT calculations. All authors contributed to the work and commented on the paper. The authors gratefully thank Ute Salzberger for the support in TEM sample preparation.

## Competing interests

The authors declare no competing interests.
