## [Peer Review File · Nature Communications]

Emergent and Robust Ferromagnetic-Insulating State in Highly Strained Ferroelastic LaCoO₃ Thin FilmsREVIEWER COMMENTS

Reviewer #1 (Remarks to the Author):

In this work, the authors reported the ordered oxygen vacancies into dark stripes and long-range suppression of CoO₆ octahedral rotations throughout the LaCoO₃ epitaxial films. Such unique structures are suggested to give rise to emergent and robust ferromagnetic-insulating state in ferroelastic LaCoO₃ films. The authors presented very high-quality STEM images and EELS spectra to clarify Co valence states and ordered oxygen vacancies in the LaCoO₃ epitaxial films. Furthermore, density functional theory calculations were conducted to reveal the mechanism and in-depth analysis to understand the stress-defect-microstructure-induced multiferroicity in LaCoO₃, which resolves long-lasting questions regarding the emergent new physics properties in the tensile-strained LaCoO₃ epitaxial films. Overall, I think this work is of board interest and import for strain multifunctional information films, especially for ferromagnetic, ferroelastic, and multiferroic materials. I like to recommend this manuscript to publish in Nature Communications with following issues/comments to be addressed.

1. The authors claimed that the present films are epitaxial films under the large tensile strain > 2.0%. Usually, for such large strain, the film will easily relax and also misfit dislocation can form at interfaces. In terms of this, a high-resolution atomic image of HAADF for the interface of LCO/STO is very necessary to illustrate whether misfit dislocation exists or not?
2. In lines of 136-137, "high-angle annular dark-field (HAADF) STEM" was repeated in the lines of 155-156. This should be corrected.
3. The authors should also include the DOS of LCO without ordered oxygen vacancies and compare with those with ordered oxygen vacancies.
4. The authors stated that they found multiple possible spin configurations with small energy difference, and hence it is not possible to uniquely determine the ground state. They should be more quantitative.

Reviewer #2 (Remarks to the Author):

The authors report a long-range suppression of CoO₆ octahedral rotations in tensile-strained LaCoO₃ epitaxial films and correlate it with the ferromagnetism and oxygen vacancies in this material by DFT calculations to explain the origin of the ferromagnetism.

The experimental conclusive evidences in this paper mainly rely on the ABF images and EELS of LaCoO₃, so ensuring the correctness of related experimental data is an important guarantee for the logic of this paper.

First, considering the scanning distortion and spatial scale calibration in STEM imaging, I suggest that the authors can supplement the ABF images of some standard samples under a same shooting condition to prove that the distortion is within an acceptable range.

Secondly, the process of fitting positions of Oxygen atoms from ABF image is also crucial for the extraction of experimental data. Therefore, I suggest the authors to add some supplementary materials in this respect, such as an ABF image with the fitted position of oxygen atom.

As for EELS, the authors claim to have observed a shift of the Co L edge position, which has not been observed in past experiments. Given that the authors used a dispersion of 0.5eV/channel and the Co L edge can shift up to 0.8-0.9eV, I suggest the authors to observe Co L edge shift with a smaller dispersion to consolidate this evidence.

In the calculation of DFT, according to the LCO diagram Fig. 4(a) provided by the author, the spatial position of Co in tetrahedron seems to change greatly, right? Does this match the experimental HADDF image?

Generally, the work in this manuscript is relatively detailed and commendable. It explains the origin of ferromagnetism in LaCoO₃ epitaxial films from a deeper perspective. If the reliability of experimental data can be guaranteed, it is believed that relevant investigate will be deeply inspired.

Reviewer #3 (Remarks to the Author):

This paper provides combined multiple experiments and theoretical calculations to find out the origin of FM insulating state in the tensile-strained LCO film grown on a SrTiO₃ substrate.

As a result, the STEM images show the formation of ordered oxygen vacancies, reducing the oxidation state of Co from 2.85 to 2.15 and promoting the ordered high-spin state of Co, which is the origin of the FM insulating state. Unfortunately, I did not see the novelty of this paper compared to other previous reports showing the FM insulating state on the LCO film grown on SrTiO₃ substrates or LaAlO₃ substrates and its origin [Nat. Commun. 12, 1843 (2021), Sci. Adv. 5, eaav5050 (2019), Phys. Rev. Lett. 112, 087202 (2014), Phys. Rev. B 91, 144418 (2015)].

In particular, ref. 14 [Phys. Rev. Lett. 112, 087202 (2014),] has already revealed that the FM insulating occurs in the tensile-strained LCO film grown on a SrTiO₃ substrate by the oxygen vacancy ordering with high-spin state Co²⁺. Additionally, ref. 14 shows the significant difference for the Co L₃/L₂ ratio in the dark and bright stripes, although there is no change in the peak positions of the Co L₂ and L₃ edge. These results are coincident with the observation in the submitted manuscript.

Therefore, I do not recommend this manuscript is not suitable for Nature Communication. Furthermore, here are some minor details the author should consider further.

1. The authors showed the formation of ordered oxygen vacancies in the 25 u.c. LCO film, but not in the 6 u.c. LCO film (Figure 1 and Figure 3). This means the formation of oxygen vacancies depends on the thickness of the LCO film. The authors should discuss the film thickness dependence on the formation of ordered oxygen vacancies. Please note this report [J. Phys. Chem. C 124, 12492-12501 (2020)].
2. Figure S1 shows that the LCO (00l) peak position moves to the lower angle with increasing film thickness, which means that the c-axis lattice parameter increase and the out-of-plane compressive strain relaxes. However, Figure 3 and Figure S6 show that the 25 u.c LCO film is still under in-plane tensile strain. The formation of oxygen vacancies ordering is close related to the strain state of an LCO film. The authors must investigate the strain state of the LCO films. It is recommended to measure reciprocal space mappings on the LCO film to discuss the strain state of the LCO films.
3. The insulating behavior of the 25 u.c. LCO film was confirmed by the experiment (Fig. S2b) and the total density of states results (Fig. 4c), although the oxygen vacancies introduced the electron to the LCO film. The authors need a detailed discussion of why the insulating behavior appears in the LCO film with the formation of the oxygen vacancies ordering.

4. XMCD measurement is a powerful tool to probe the spin state of Co ion with ferromagnetism directly. So, the authors performed the XMCD measurement at Co-L_{2,3} edges on the 25 u.c. LCO film to reveal that FM ordering originates from Co ion in the film, as shown in Figure S2a. However, the XMCD results in this manuscript are very poor at showing the change in the spin state of Co ions in the LCO film by the oxygen vacancies ordering. As Fig. 6 shows, to investigate the spin state of Co ion by the formation of oxygen vacancies ordering, the authors have to need to compare the XMCD results of the 25 u.c. LCO film containing the oxygen vacancies ordering and that of the 6 u.c. LCO film without oxygen vacancies ordering. Please note the reported paper [Phys. Rev. B 91, 144418 (2015)].

Response to Reviewers' Comments

We appreciate the reviewers' time and effort for reviewing our manuscript. The reviewers' comments and suggestions were very constructive and helped us to improving our manuscript. The following is point-to-point response to their comments. We have properly addressed all the questions from the reviewers and made all necessary changes in the revised manuscript. All the revisions in the revised manuscript are highlighted in yellow.

Reviewer #1

In this work, the authors reported the ordered oxygen vacancies into dark stripes and long-range suppression of CoO_6 octahedral rotations throughout the LaCoO_3 epitaxial films. Such unique structures are suggested to give rise to emergent and robust ferromagnetic-insulating state in ferroelastic LaCoO_3 films. The authors presented very high-quality STEM images and EELS spectra to clarify Co valence states and ordered oxygen vacancies in the LaCoO_3 epitaxial films. Furthermore, density functional theory calculations were conducted to reveal the mechanism and in-depth analysis to understand the stress-defect-microstructure-induced multiferroicity in LaCoO_3 , which resolves long-lasting questions regarding the emergent new physics properties in the tensile-strained LaCoO_3 epitaxial films. Overall, I think this work is of broad interest and important for strain multifunctional information films, especially for ferromagnetic, ferroelastic, and multiferroic materials. I like to recommend this manuscript to publish in Nature Communications with following issues/comments to be addressed.

We thank the reviewer for the thorough review and for considering that it “*resolves long-lasting questions* regarding the emergent new physics properties in the tensile-strained LaCoO_3 epitaxial films”, and believing that “this work is of *broad interest and important* for strain multifunctional information films, especially for ferromagnetic, ferroelastic, and multiferroic materials”, and also recommending “this manuscript to publish in Nature Communications”. The following are point-by-point responses to the reviewer's comments.

1. *The authors claimed that the present films are epitaxial films under the large tensile strain > 2.0%. Usually, for such large strain, the film will easily relax and also misfit dislocation can form at interfaces. In terms of this, a high-resolution atomic image of HAADF for the interface of LCO/STO is very necessary to illustrate whether misfit dislocation exists or not?*

Response:

We thank the reviewer for this valuable suggestion. We fully agree with the reviewer that for epitaxial thin films with large lattice mismatch between the film and substrate, misfit dislocations may form at the interface to relax the strain when the epilayer is thick. Nevertheless, this phenomenon does not occur in LaCoO_3 thin films. According to the detailed studies in our studies and literature, we found that LaCoO_3 thin films are prone

to form dark strips to relieve the large epitaxial strain rather than forming misfit dislocations at the interface between LaCoO_3 and SrTiO_3 . To further clarify whether misfit dislocation exists or not, as shown in Supplementary Figure S7, we have carried out inverse FFT analysis of STEM results on the interface. It clearly shows that the atomic planes across the interface are coherent, revealing that there is no misfit dislocations existing at the interface.

Supplementary Figure S7 (a) The atomic resolution HAADF-STEM image of the interface between LCO and STO; the inset is the FFT image. (b) the corresponding inverse FFT image using the reflections marked by red circles in the inset of (a).

Action taken:

In the revised manuscript, Figure S7 has been added to the Supplementary Information. We have also added the following discussions on pages 6-7: “Fig. 1i displays the EELS line profiles of individual elements of 25-uc-thick LCO film, revealing an abrupt interface between LCO and STO without the formation of any misfit dislocations (Supplementary Fig. 7).”.

2. In lines of 136-137, “high-angle annular dark-field (HAADF) STEM” was repeated in the lines of 155-156. This should be corrected.

Response:

We thank the reviewer for the careful review. In the revised manuscript, we have removed the repeated sentence on page 7: “An EELS line scan measured from an atomic-resolution HAADF-STEM image of 25-uc-thick LCO film is shown in Fig. 2b and a

periodic reduction of the intensity of the Co- $L_{2,3}$ edges is observed, coinciding with the positions of the observed dark stripes.”.

3. The authors should also include the DOS of LCO without ordered oxygen vacancies and compare with those with ordered oxygen vacancies.

Response:

We thank the reviewer for pointing this out. Following the reviewer’s suggestion, as shown in Supplementary Figure S12, we have further calculated the DOS of LaCoO_3 without ordered oxygen vacancies, *i.e.* the pristine bulk LaCoO_3 . The checkerboard HS-LS spin configuration was found to give the lowest energy, where half of the Co ions are in the high-spin (HS) state and others are in the low-spin (LS) state. The conduction band minimum is made of the t_{2g} state of the HS Co^{3+} , while the valence band maximum consists of the t_{2g} state of the LS Co^{3+} , and the band gap was found to be about 0.77 eV. In comparison, the DOS of LaCoO_3 with ordered oxygen vacancies (Figure 5b) shows its valence band maximum also consists mainly of Co t_{2g} states, although the assignment of the conduction band minimum is difficult due to the existence of Co in multiple oxidation and coordination states. Nevertheless, the band gap is still formed by the Co 3d states, which is consistent with the observed insulating state. The extra electrons resulting from oxygen vacancies only lead to the reduction of Co^{3+} to Co^{2+} .

Supplementary Figure S12 Projected density of state (PDOS) spectrum for (a) LCO without ordered V_O or dark stripes, (b) Co ions with low-spin (LS) states, and (c) Co ions with high-spin (HS) states.

Action taken:

In the revised manuscript, Figure S12 has been added to the Supplementary Information. We have also added the following discussions on page 11: “The HS-LS checkerboard spin configuration in the pristine LCO without ordered V_O was found to have the lowest energy with a band gap opening between the t_{2g} states of HS and LS Co^{3+} (Supplementary Fig.12). The presence of V_O would result in conducting states as a result of the extra electrons. This is not the case here as the extra electrons lead to the reduction of Co^{3+} to Co^{2+} , while the nature of the band gap remains unchanged, which is between the Co 3d states caused by the crystal-field splitting.”.

4. *The authors stated that they found multiple possible spin configurations with small energy difference, and hence it is not possible to uniquely determine the ground state. They should be more quantitative.*

Response:

We would like to thank the reviewer’s valuable suggestion. For enumeration, one could initialize the spin configuration as +3/-3 for the HS Co^{2+} sites. As there are eight such sites this would require $2^8/2 = 128$ relaxations each containing 56 atoms. The problem becomes more complicated when the HS/LS Co sites are allowed to change. Hence, we believe it is not necessary to enumerate all configurations. Overall, we have found configurations with a total spin of 0, 6, 12, 24 (per 12 Co atoms) with energy differences up to 30 meV per atom, and the lowest energy configurations with a total spin of 0, 6, 24 are only a few meV per atom difference in energy. On the other hand, a configuration with a full FM spin configuration including high spin Co^{3+} results in high energies of more than 50 meV per atom. We note that the relative energies are also likely to be affected by the value of U_{eff} applied to Co atoms. As a result, we believe it is not possible to unambiguously determine the magnetic spin state based on the DFT calculations.

Action taken:

In the revised manuscript, we have added the following discussions on pages 14-15: “The fully FM case with HS Co^{3+} (total spin $20 \mu_B$) is found to result in a much higher energy, up to more than 50 meV per atom, while multiple arrangements with a total spin of $12 \mu_B$, $6 \mu_B$ or $0 \mu_B$ are found to give lower energies. Due to the small difference in energies (in the order of a few meV per atom), a unique ground state spin arrangement cannot be determined reliably using DFT.”.

Reviewer #2

The authors report a long-range suppression of CoO_6 octahedral rotations in tensile-strained $LaCoO_3$ epitaxial films and correlate it with the ferromagnetism and oxygen

vacancies in this material by DFT calculations to explain the origin of the ferromagnetism.

The experimental conclusive evidences in this paper mainly rely on the ABF images and EELS of LaCoO₃, so ensuring the correctness of related experimental data is an important guarantee for the logic of this paper.

We greatly appreciate the reviewer for your careful review and valuable comments for improving our manuscript. We fully agree with the reviewer that the STEM and EELS data are crucial for the conclusion of our work. We addressed all the comments and suggestions as following.

1. *First, considering the scanning distortion and spatial scale calibration in STEM imaging, I suggest that the authors can supplement the ABF images of some standard samples under a same shooting condition to prove that the distortion is within an acceptable range?*

Response:

We thank the reviewer for the valuable suggestion. We have supplemented some STEM images of the standard sample SrTiO₃ (*i.e.*, SrTiO₃ substrate used for growing LaCoO₃ thin film in our paper) under the same imaging condition. The atomic position of oxygen ions can be clearly visualized (Supplementary Figure S9), showing non-distorted oxygen octahedra along the viewing direction (100), which is in good agreement with the cubic crystal structure ($a^0a^0a^0$) of SrTiO₃. These results demonstrate that the effects of scanning distortion are very limited and, thus, negligible in this work.

Supplementary Figure S9 Simultaneously acquired HAADF (a) and ABF (b) images of the standard SrTiO₃. (c) The corresponding ABF image (b) with fitted oxygen (red circles) and Ti positions (yellow circles).

Action taken:

In the revised manuscript, Figure S9 has been added to the Supplementary Information. We have also added the following discussions on page 9: “Annular bright-field (ABF) STEM imaging^{8,22,30} was further applied to quantitatively measure the cation and anion positions (Supplementary Fig. 9 and Supplementary Fig. 10) for determining the Co-O-

Co bond angle ($\beta_{Co-O-Co}$) and the Co-O bond length (d_{Co-O})." and "According to the Glazer notation^{31,32}, the TiO_6 rotation pattern in the STO substrate is confirmed as $a^0a^0a^0$ (Supplementary Fig. 9) and the rotation pattern of LCO is determined to be $a^-a^-c^-$ or $a^-a^-c^+$ (Supplementary Fig. 10), consistent with $a^-a^-a^-$ of bulk LCO²⁰."

2. Secondly, the process of fitting positions of Oxygen atoms from ABF image is also crucial for the extraction of experimental data. Therefore, I suggest the authors to add some supplementary materials in this respect, such as an ABF image with the fitted position of oxygen atom.

Response:

We fully agree with the reviewer’s suggestion. In our response to the previous comment, we have supplemented ABF images of SrTiO₃ with the fitted position of oxygen atoms (Supplementary Figure S9). Furthermore, ABF images of LaCoO₃ thin film (Supplementary Figure S10) with the fitted position of oxygen atoms are also added, evidencing the high-precision fitting of the oxygen atoms in this work.

Supplementary Figure S10 Inverted ABF image of 6 uc LCO (a) and 25 uc LCO with fitted position of oxygen ions (red circles) and Co (Ti) ions (yellow circles).

Action taken:

In the revised manuscript, Figure S10 has been added to the Supplementary Information. We have also added the following discussions on page 9: "Annular bright-field (ABF) STEM imaging^{8,22,30} was further applied to quantitatively measure the cation and anion positions (Supplementary Fig. 9 and Supplementary Fig. 10) for determining the Co-O-Co bond angle ($\beta_{Co-O-Co}$) and the Co-O bond length (d_{Co-O})." and "According to the Glazer notation^{31,32}, the TiO_6 rotation pattern in the STO substrate is confirmed as $a^0a^0a^0$

(Supplementary Fig. 9) and the rotation pattern of LCO is determined to be $a^-a^-c^-$ or $a^-a^-c^+$ (Supplementary Fig. 10), consistent with $a^-a^-a^-$ of bulk LCO²⁰.”.

3. As for EELS, the authors claim to have observed a shift of the Co L edge position, which has not been observed in past experiments. Given that the authors used a dispersion of 0.5eV/channel and the Co L edge can shift up to 0.8-0.9eV, I suggest the authors to observe Co L edge shift with a smaller dispersion to consolidate this evidence.

Response:

We thank the reviewer for pointing out this. We did not provide the experimental details for EELS studies in the previous version. We use a dispersion of 0.5 eV/channel for EELS 2D elemental mapping for the simultaneous acquisition of signals of Ti- $L_{2,3}$, O-K, Co- $L_{2,3}$, La- $M_{4,5}$, and Sr- $L_{2,3}$ edges with a Gatan K2 Summit camera. For analyzing the fine structure of O-K and Co- $L_{2,3}$ edges, we used a dispersion of 0.1 eV/channel to acquire the presented EELS line scan results (Figure 2) with a CCD camera in the Dual-EELS mode. In this case, as shown in Supplementary Figure S14, the energy resolution is about 0.6 eV, consolidating the observed shift of the Co-L edge position.

Supplementary Figure S14 Simultaneously acquired zero-loss spectrum during EELS line scan (Figure 2). The estimated energy resolution is about 0.6 eV.

Action taken:

In the revised manuscript, Figure S14 has been added to the Supplementary Information. We have also supplemented the experimental details for our EELS studies in the method section on page 17: “EELS acquisition has been performed with a Gatan GIF Quantum ERS imaging filter equipped with a Gatan K2 Summit camera and a CCD camera with a convergent semi-angle of 20.4 mrad and a collection semi-angle of 111 mrad. EELS spectrum imaging was performed with a dispersion of 0.5 eV/channel for the

simultaneous acquisition of signals of the Ti- $L_{2,3}$, O-K, Co- $L_{2,3}$, La- $M_{4,5}$, and Sr- $L_{2,3}$ edges. EELS line scans were conducted with a dispersion of 0.1 eV/channel using a CCD camera in the Dual-EELS mode for further fine structure analysis of the O-K and Co- $L_{2,3}$ edges. The corresponding energy resolution is about 0.6 eV (Supplementary Fig. 14).”.

4. In the calculation of DFT, according to the LCO diagram Fig. 4(a) provided by the author, the spatial position of Co in tetrahedron seems to change greatly, right? Does this match the experimental HAADF image?

Response:

We thank the reviewer for raising this question. The DFT calculations adopt models constructed using the low energy structures found by a random structure search. Indeed, we found that the Co atoms are displaced from the initial octahedral positions. For simplicity, Figure 4(a) *only* shows *a single layer* of LaCoO_3 . However, when the unit cell is repeated along the b and c lattice vectors, as shown in Figure R1, the tetrahedral Co appears to have a *zigzag displacement* from the octahedral locations in the same viewing direction. Such displacement of Co atoms would average out in the HAADF image along the viewing direction, which is consistent with the reduced contrast in the dark regions (in other words, this matches the experimental HAADF image). In addition, potential site-occupation disordering of oxygen vacancies in the dark region could further reduce the apparent off-site displacements of the Co atoms.

Figure R1 Schematic view of LCO with ordered V_{O} and with repeated unit cell along the b and c lattice vectors.

Action taken:

In the revised manuscript, we have added the following discussions on page 11: “The tetrahedral coordinated Co atoms are displaced from their ideal octahedral positions, resulting in a zigzag arrangement which makes the column of atoms no longer aligned when viewed along the same direction as in Fig. 4a. We note that for clarity, Fig. 4a only includes a single layer of LCO which does not show this effect. Such misaligned atomic

columns could also be one possible origin of the reduced contrast in the dark region of the HAADF image (Fig. 1c).”.

5. Generally, the work in this manuscript is relatively detailed and commendable. It explains the origin of ferromagnetism in LaCoO_3 epitaxial films from a deeper perspective. If the reliability of experimental data can be guaranteed, it is believed that relevant investigate will be deeply inspired.

Response:

We very much appreciate the reviewer for the careful review of our manuscript and very positive feedback.

Reviewer #3

This paper provides combined multiple experiments and theoretical calculations to find out the origin of FM insulating state in the tensile-strained LCO film grown on a SrTiO_3 substrate.

As a result, the STEM images show the formation of ordered oxygen vacancies, reducing the oxidation state of Co from 2.85 to 2.15 and promoting the ordered high-spin state of Co, which is the origin of the FM insulating state. Unfortunately, I did not see the novelty of this paper compared to other previous reports showing the FM insulating state on the LCO film grown on SrTiO_3 substrates or LaAlO_3 substrates and its origin [Nat. Commun. 12, 1853 (2021), Sci. Adv. 5, eaav5050 (2019), Phys. Rev. Lett. 112, 087202 (2014), Phys. Rev. B 91, 144418 (2015)].

We thank the reviewer for the thorough review and critical comments. We agree with the reviewer that there are a few previous works [Nat. Commun. 12, 1853 (2021), Sci. Adv. 5, eaav5050 (2019), Phys. Rev. Lett. 112, 087202 (2014), Phys. Rev. B 91, 144418 (2015)] reporting the FM insulating state in LaCoO_3 thin films grown on SrTiO_3 or LaAlO_3 substrates. However, we would like to emphasize that the *origin* of the long-range FM order observed in tensile-strained LaCoO_3 thin films remains *unclear* and *controversial*. Please find a brief summary below (to clarify the novelty of our paper):

- (1) In the mentioned reference [Nat. Commun. 12, 1853 (2021)], the FM insulating state is found in a metastable $\text{LaCoO}_{2.5}$ phase formed by annealing a compressive-strained LaCoO_3 thin film (grown on LaAlO_3 substrate) in the vacuum. The origin of long-range FM order observed in the $\text{LaCoO}_{2.5}$ phase is due to the formation of ordered oxygen vacancies *via* reducing LaCoO_3 thin film in vacuum. We want to point out that *the $\text{LaCoO}_{2.5}$ phase is totally different from the LaCoO_3 phase investigated in our work*. In addition, it can be seen that the formation of ordered oxygen vacancies is responsible for inducing the FM insulating state.

- (2) In the mentioned reference [Sci. Adv. 5, eaav5050 (2019)], the tensile-strain induced *ferroelastic distortion* was proposed as the driving force for inducing FM order in LaCoO₃ thin film. This mechanism has been used for explaining the origin of long-range FM order observed in a LaCoO₃ thin film [Nano Lett. 12, 4966-4970 (2012), Phys. Rev. Lett. 122, 187202 (2019)], as we stated in the main text.
- (3) In the mentioned references [Phys. Rev. Lett. 112, 087202 (2014), Phys. Rev. B 91, 144418 (2015)], the origin of long-range FM order observed in LaCoO₃ thin film was attributed to *the formation of ordered oxygen vacancies*. This mechanism is another one which has been used for explaining the origin of long-range FM order observed in tensile-strained LaCoO₃ thin film, as we cited these references as refs.14 and 15 in the main text.

Based on above summary, we can find that the *exact origin* of the long-range FM order observed in tensile-strained LaCoO₃ thin films remains *unclear* and *controversial*. Here, in our paper, by using a combination of advanced characterizations (include STEM, EELS, and spectroscopy) and DFT calculations:

- (1) We simultaneously revealed the formation of ordered oxygen vacancies (Figure 2) and previously *unreported long-range suppression of CoO₆ octahedral rotations (up to 25 uc)* as shown in Figure 3) throughout tensile-strained LaCoO₃ thin films.
- (2) We further identified that the *dual effect* (the formation of ordered oxygen vacancies and long-range suppression of CoO₆ octahedral rotations) strongly modifies the Co 3d-O 2p hybridization (Figure 2 and Figure 4) associated with the increase of both Co-O-Co bond angle and Co-O bond length (Figure 3) which in turn induces long-range FM order in tensile-strained LaCoO₃ thin films.
- (3) This *dual effect* is *a novel physical mechanism* (totally different from previously reported ferroelastic distortion or the formation of ordered oxygen vacancies) that can be used for resolving the *long-standing questions* on *the exact origin* of the long-range FM order observed in the tensile-strained LaCoO₃ thin films.

Action taken:

In the revised manuscript, in order to clarify the novelty of our work more clearly, we have modified the sentence in the abstract on page 2: “Here, by precisely determining atomic and electronic structures in tensile-strained LaCoO₃ epitaxial films, we simultaneously reveal the formation of ordered oxygen vacancies and previously unreported long-range (up to 25 unit cells) suppression of CoO₆ octahedral rotations throughout the films.”, cited the reference [Nat. Commun. 12, 1853 (2021), Sci. Adv. 5, eaav5050 (2019)] mentioned by the reviewer, added one sentence on page 3: “By annealing a compressive-strained LCO epitaxial film in vacuum, the LaCoO_{2.5} phase with

a zigzag-like oxygen vacancies (V_O) ordering shows a ferromagnetism (FM) insulating state^{11.}”, changed the sentence on page 3: “Previously, the tensile-strain induced ferroelastic distortion was proposed as the driving force for inducing FM order^{14,17,18.}”, also modified the sentence in the conclusion on page 15-16: “Consequently, owing to the dual effect of the formation of ordered V_O and long-range suppressed CoO_6 octahedral rotations, the crystal-field splitting is weakened, thus promoting the ordered high-spin state of Co ions, and producing an emergent and robust ferromagnetic insulating state.”.

In particular, ref. 14 [Phys. Rev. Lett. 112, 087202 (2014),] has already revealed that the FM insulating occurs in the tensile-strained LCO film grown on a $SrTiO_3$ substrate by the oxygen vacancy ordering with high-spin state Co^{2+} . Additionally, ref. 14 shows the significant difference for the Co L_3/L_2 ratio in the dark and bright stripes, although there is no change in the peak positions of the Co L_2 and L_3 edge. These results are coincident with the observation in the submitted manuscript.

Response:

We thank the reviewer for this comment. Regarding the Co EELS data shown in ref. 14 [Phys. Rev. Lett. 112, 087202 (2014)], we agree with the reviewer that our conclusion that based on EELS results mostly coincides with the previous work. Nevertheless, we present more convincing studies, providing concrete evidence for settling the controversy over the origin of FM order in $LaCoO_3$ epitaxial films: 1). The variation of the O-K pre-peak, the Co L_3/L_2 ratio and the Co- L_3 edge position generally fingerprint the Co valence change. EELS analysis in ref. 14 *only* observes the difference for the O-K pre-peak and the Co L_3/L_2 ratio in the dark and bright stripes, but not for the Co- L_3 edge position. *This may raise a concern that the Co has different spin states but not necessarily a differing valance state.* Our work shows the coexistence of the difference for the Co L_3/L_2 ratio and the Co- L_3 edge position in the dark and bright stripes, enhancing the argument about the Co valence change. The improvement might result from our EELS measurement with a good energy resolution (0.6 eV in Supplementary Figure S14); thus, a shift of 0.8-0.9 eV for the Co- L_3 edge can be resolved. 2) Unlike ref. 14, which *only* shows a single EELS spectrum from the dark and bright stripes respectively, we present an EELS line scan across the dark and bright stripes (ca. 8 nm), showing the regular shift of the Co L_3/L_2 edges to lower energy at each dark stripe (Figure 2b), directly evidencing the long-range ordering of the Co valence change.

Therefore, I do not recommend this manuscript is not suitable for Nature Communication.

Based on the detailed discussions above, we believe the novelty of our paper has been convincingly demonstrated compared to other previous reports mentioned by the reviewer, and we trust that the reviewer will be convinced of this.

Furthermore, here are some minor details the author should consider further.

1. *The authors showed the formation of ordered oxygen vacancies in the 25 u.c. LCO film, but not in the 6 u.c. LCO film (Figure 1 and Figure 3). This means the formation of oxygen vacancies depends on the thickness of the LCO film. The authors should discuss the film thickness dependence on the formation of ordered oxygen vacancies. Please note this report [J. Phys. Chem. C 124, 12492-12501 (2020)].*

Response:

We thank the reviewer for this valuable suggestion and fully agree with the reviewer that the formation of oxygen vacancies strongly depends on the thickness of the LaCoO₃ thin film, which is consistent with the observation reported in the reference [J. Phys. Chem. C 124, 12492-12501, (2020)] mentioned by the reviewer.

Action taken:

In the revised manuscript, we have cited the reference [J. Phys. Chem. C 124, 12492-12501, (2020)] mentioned by the reviewer and also added the following discussions on page 8: “These results strongly imply that the formation of ordered V_O is gradually developed in LCO films with increasing thickness, consistent with a previous report²⁹, and further corroborating that in 25-uc-thick LCO film the formation of dark stripes or ordered V_O are strongly coupled with emergent FM order.”.

2. *Figure S1 shows that the LCO (00l) peak position moves to the lower angle with increasing film thickness, which means that the c-axis lattice parameter increase and the out-of-plane compressive strain relaxes. However, Figure 3 and Figure S6 show that the 25 u.c LCO film is still under in-plane tensile strain. The formation of oxygen vacancies ordering is close related to the strain state of an LCO film. The authors must investigate the strain state of the LCO films. It is recommended to measure reciprocal space mappings on the LCO film to discuss the strain state of the LCO films.*

Response:

We thank the reviewer for this excellent suggestion. To investigate the strain state, as shown in Supplementary Figure S2 we have further carried out high-resolution asymmetric X-ray reciprocal space maps (RSMs) for LaCoO₃ films with thicknesses of 6 uc, 8 uc, 12 uc, and 25 uc. It can be seen that (103) peaks of SrTiO₃ and LaCoO₃ have the same q_x, indicating that LaCoO₃ thin films are fully tensile strained to the SrTiO₃ substrate along the in-plane direction. This is consistent with strain results from Figure 3 and Figure S11.

Supplementary Figure S2 Reciprocal space maps of (103) Bragg reflections of STO and LCO for LCO thin films with different thickness.

Action taken:

In the revised manuscript, Figure S2 has been added to the Supplementary Information. We have also added the following discussion on page 6: “Reciprocal space maps of (103) Bragg reflections of STO and LCO exhibited that LCO thin films are fully tensile strained to the STO substrates along the in-plane direction (Supplementary Fig. 2).”.

3. *The insulating behavior of the 25 u.c. LCO film was confirmed by the experiment (Fig. S2b) and the total density of states results (Fig. 4c), although the oxygen vacancies introduced the electron to the LCO film. The authors need a detailed discussion of why the insulating behavior appears in the LCO film with the formation of the oxygen vacancies ordering.*

Response:

We thank the reviewer for the comment. For metal oxides such as In_2O_3 , Ga_2O_3 , SnO_2 , ZnO , TiO_2 which are n-type semiconductors, oxygen vacancies are n-type dopants which donate extra electrons into the conduction band and result in electrical conductivity. However, for many transition metal oxides such as LaCoO_3 , Co_3O_4 , LaMnO_3 , the transition metal cations (*e.g.*, Co) have multiple oxidation states. The band gap is formed by filled and empty states of transition metal 3*d* states (hybridized with O 2*p* to some extent). The extra electrons caused by the oxygen vacancies will be localized at the Co 3*d* orbital by reducing some Co cations from +3 oxidation state to +2 state instead of forming free electrons for electrical conductivity.

Action taken:

In the revised manuscript, we have added the following discussions on page 11: “The presence of V_O would result in conducting states as a result of the extra electrons. This is not the case here as the extra electrons lead to the reduction of Co^{3+} to Co^{2+} , while the nature of the band gap remains unchanged, which is between the Co 3d states caused by the crystal-field splitting.”.

4. XMCD measurement is a powerful tool to probe the spin state of Co ion with ferromagnetism directly. So, the authors performed the XMCD measurement at Co-L2,3 edges on the 25 u.c. LCO film to reveal that FM ordering originates from Co ion in the film, as shown in Figure S2a. However, the XMCD results in this manuscript are very poor at showing the change in the spin state of Co ions in the LCO film by the oxygen vacancies ordering. As Fig. 6 shows, to investigate the spin state of Co ion by the formation of oxygen vacancies ordering, the authors have to need to compare the XMCD results of the 25 u.c. LCO film containing the oxygen vacancies ordering and that of the 6 u.c. LCO film without oxygen vacancies ordering. Please note the reported paper [Phys. Rev. B 91, 144418 (2015)].

Response:

We thank the reviewer for the constructive comments. By reading the reference [Phys. Rev. B 91, 144418 (2015)] suggested by the reviewer, we speculate the reviewer wanted us to refer to the XAS and XMCD spectra shown in Figure 5 and Figure 9 (as shown in Figure R2), respectively, instead of Figure 6 in the reference.

Figure R2 XAS and XMCD spectra of $LaCoO_3$ thin films grown on $Nb-SrTiO_3$ substrates measured at 25 K [Adapted from Figure 5 and Figure 9 presented in Phys. Rev. B 91, 144418 (2015), respectively].

Supplementary Figure S3 (a) XAS spectra (top panel) for LCO 25 uc and 6 uc, and reference spectra (bottom panel) for Co^{2+} HS, Co^{3+} HS, and Co^{3+} LS. (b) XMCD spectra for LCO 25 uc and 6 uc.

Following the reviewer's suggestions and referring to the XAS and XMCD spectra shown in Phys. Rev. B 91, 144418 (2015), as shown in Supplementary Figure S3(a), we have compared the XAS and XMCD spectra measured from the 25 uc LaCoO_3 thin film with those of the 6 uc LaCoO_3 thin film. Compared to our Co reference spectra and XAS spectra shown in Figure R2 [Phys. Rev. B 91, 144418 (2015)], we indeed observe a similar change in the spin state of Co ions in the 25 uc LaCoO_3 thin film by the formation of ordered oxygen vacancies. Both of 6 uc and 25 uc LaCoO_3 thin films are in the mixed high-spin (HS) and low-spin (LS) states. We find that the 25 uc LaCoO_3 thin film (with ordered oxygen vacancies) has a larger HS contribution compared to the 6 uc LaCoO_3 thin film (without ordered oxygen vacancies). Consequently, as shown in Supplementary Figure S3(b), the 6 uc LaCoO_3 thin film (without ordered oxygen vacancies) shows weak magnetism compared to 25 uc LaCoO_3 thin film (with ordered oxygen vacancies), which is consistent with the macroscopic magnetic result shown in Figure 1a (main text).

Action taken:

In the revised manuscript, Figure S3 has been added to the Supplementary Information. We have also added the following discussions on page 6: “X-ray magnetic circular dichroism (XMCD) and XAS measurements at Co- $L_{2,3}$ edges further revealed that FM originates from Co ion in the film and the change in the spin states of Co ions in the films is consistent with the formation of ordered V_O^{16} (Supplementary Fig. 3).”.

REVIEWERS' COMMENTS

Reviewer #1 (Remarks to the Author):

The authors have responded to all my comments and in view of the response to other comments, in my opinion the revised version has addressed my concerns. With these, I recommend the publication of the manuscript in Nature Communications.

Reviewer #2 (Remarks to the Author):

The authors have carefully revised the manuscript, resulting in an improved quality of the manuscript. There still are two questions about this article:

1. As we know, the lattice constant for STO is about 3.9 Å, but in the Supplementary Figure S9, this constant is obviously larger than 5 Å if the corresponding scale bar is correct. I suggest the authors explain for that and amend this picture.
2. According to the SI of Nano Lett. 21, 4006 (2021), the ordered structure in LaCoO₃ is sensitive to electron dose, i.e., “Interestingly, in the high-dose condition, the Co–L3 edge of a dark stripe is red-shifted from that of a bright unit (here, b-unit). Meanwhile, in the low-dose condition, the positions of Co–L3 edge are identical in all regions including dark stripes and bright units” (copied from the reference). It is recommended to inspect the effect of electron dose to experimental results.

Response to Reviewers' Comments

We appreciate the reviewers' time and effort in reviewing our re-submission and are very happy to know that all reviewers are satisfying with our previous revisions. We have properly addressed the new comments from the reviewer #2 and made all necessary changes in the revised manuscript. All the revisions in the revised manuscript are highlighted in yellow.

Reviewer #1

The authors have responded to all my comments and in view of the response to other comments, in my opinion the revised version has addressed my concerns. With these, I recommend the publication of the manuscript in Nature Communications.

We thank the reviewer for recommending the publication of our manuscript in Nature Communications without further revision.

Reviewer #2

The authors have carefully revised the manuscript, resulting in an improved quality of the manuscript.

We are very happy to know that the reviewer is satisfying with our previous revisions. We addressed the new comments as following.

There still are two questions about this article:

1. As we know, the lattice constant for STO is about 3.9 Å, but in the Supplementary Figure S9, this constant is obvious larger than 5 Å if the corresponding scale bar is correct. I suggest the authors explain for that and amend this picture.

Response:

We thank the reviewer for pointing out this. There was an error for the scale bar in the previous version. We have now corrected the scale bar in the updated **Supplementary Figure S9**, as shown below.

Supplementary Figure S9 STEM investigation of oxygen octahedra in STO. Simultaneously acquired HAADF (a) and ABF (b) images of the standard STO. (c) The

corresponding ABF image (b) with fitted oxygen (red circles) and Ti positions (yellow circles).

2. According to the SI of *Nano Lett.* 21, 4006 (2021), the ordered structure in LaCoO_3 is sensitive to electron dose, i.e., “Interestingly, in the high-dose condition, the Co- L_3 edge of a dark stripe is red-shifted from that of a bright unit (here, b-unit). Meanwhile, in the low-dose condition, the positions of Co- L_3 edge are identical in all regions including dark stripes and bright units” (copied from the reference). It is recommended to inspect the effect of electron dose to experimental results.

Response:

We thank the reviewer for the valuable suggestion. Following the reviewer’s suggestion, we have further conducted additional EELS measurements, as shown in **Supplementary Figure S15**. It clearly shows that the Co- L_3 edge of a dark strip is red-shifted from that of a bright strip in our sample, regardless of the electron dose used in the EELS measurements. This demonstrates that our findings are intrinsic for our sample rather than induced by the electron beam.

Supplementary Figure S15 EELS investigation with different electron dose conditions. Co- $L_{2,3}$ spectra of dark stripes and bright stripes measured with (a) high- and (b) low-dose conditions in 25 uc LCO. For the high-dose condition, the EELS spectrum was obtained by exposing for a long time at a specific position (25 sec/1 point). For comparison, a low-dose EELS spectrum was achieved by integrating short-exposure-time spectra obtained at equivalent but various points (0.1 sec/1 point \times 20 points).

Action taken:

In the revised manuscript, Figure S15 has been added to the Supplementary Information. Also, we have cited the reference [*Nano Lett.* 12, 4006 (2021)] and added the following discussions on page 17: “The effect of electron dose conditions on EELS measurements was evaluated according to the literature⁴⁸, showing that the observed red shift of Co- L_3 edge in dark stripes compared to bright stripes is not caused by the electron dose (Supplementary Fig. S15)”.